# Characteristics and Therapy of Jersey Finger Type V Injuries at a Middle-European Level 1 Trauma Center—A Retrospective Data Analysis

**DOI:** 10.3390/jcm13216540

**Published:** 2024-10-31

**Authors:** Paul Lennart Hoppe, Stephan Frenzel, Irena Krusche-Mandl, Gerhild Thalhammer, Stefan Hajdu, Gabriel Halát

**Affiliations:** Department of Orthopedics and Trauma Surgery, Medical University of Vienna, Währinger Gürtel 18-21, 1090 Vienna, Austria; stephan.frenzel@meduniwien.ac.at (S.F.); irena.krusche-mandl@meduniwien.ac.at (I.K.-M.); gerhild.thalhammer@meduniwien.ac.at (G.T.); stefan.hajdu@meduniwien.ac.at (S.H.); gabriel.halat@meduniwien.ac.at (G.H.)

**Keywords:** jersey finger, flexor digitorum profundus avulsion, bony avulsion, hand surgery

## Abstract

**Background/Objectives**: Reports on type V FDP tendon avulsions and their treatment are rare. Furthermore, they are not always classified in a consistent manner in the literature. The purpose of our retrospective data analysis was to evaluate and present jersey finger type V injury characteristics, primary radiological findings, treatment options and subsequent patient outcomes, as well as potential complications. **Methods**: We reviewed all patients treated for a fracture of the distal phalanx at an academic Level 1 trauma center over a period of 19 years. By reviewing the patients’ charts and their initial X-rays, we identified 44 patients with injuries matching the criteria for classification as jersey finger type Va and type Vb. All clinical records and radiologic images were reviewed to gather data on the mechanism of trauma, injury characteristics, type of treatment and subsequent outcomes in both subtypes. **Results**: Direct blows represented the most common mechanism of trauma, accounting for 23 cases. Among 44 jersey finger type V injuries, 31 showed minor displacement and were treated conservatively with a good outcome. Six patients undergoing surgery showed a poor outcome, except for one. **Conclusions**: Jersey finger type V differs considerably from the remaining types of jersey finger injuries regarding the predominant trauma mechanism. Therefore, its inclusion in this classification should be reevaluated. Established surgical techniques for refixation did not show a satisfying outcome, thus the implementation of alternative surgical techniques seems advisable when better therapeutic results are sought.

## 1. Introduction

Jersey finger is defined as a traumatic avulsion of the flexor digitorum profundus (FDP) tendon. It is named after the most common mechanism of injury, as the injury occurs in rugby players when grasping the jersey of a moving opponent. In this situation, the active flexion of the player’s finger coincides with the passive extension resulting from the movement of the opponent, overstressing the site of insertion of the FDP tendon. Due to anatomical and biomechanical characteristics, it affects the ring finger in 75% of cases [1].

Leddy and Packer were the first to classify this injury into three types in 1977 [1,2,3]. Type I and type II were described as tendon avulsions, with or without a small bony flake, in which the tendon either retracts into the palm (type I) or to the level of the proximal interphalangeal joint (type II). Type III was defined as a major bony avulsion, which is not retracted beyond the A4 pulley. Robins and Smith [4,5] added a jersey finger type IV to this classification, which includes not only a bony avulsion, but an additional avulsion of the tendon from the bony fragment.

Jersey finger type V was described by Al-Quattan in 2001 [6]. He defined it as an FDP tendon avulsion with a concurrent transverse fracture of the distal phalanx, resulting in an extraarticular (type Va) or an intraarticular avulsion fracture (type Vb) (Figure 1). This type of jersey finger is deemed to be the least frequent. In the literature, only case reports or case series with a maximum of four patients can be found [7,8,9]. In many cases, they are not even classified as jersey finger type V, which indicates the controversial understanding of this injury [10,11].

Likewise, the published treatment options are diverse, depending on the preferences of the authors. In all published cases, severe displacement was observed, so every patient underwent surgery. For refixation, axial K-wire stabilization and tendon grasping pull-out sutures were utilized. Use of a miniplate and sutures fixating the tendon to the plate was described as well. In all reported jersey finger type V repair techniques, complications are common, either due to infections and nail growth disorders in extracutaneous fixations, or tactile irritation and foreign body sensation due to material load in internal fixations [12,13,14]. Recent studies have reported high rates of complications in patients with FDP tendon repair in zone 1, in terms of osteomyelitis, wound dehiscence, chronic draining granuloma and the need for secondary surgeries, challenging the benefits of surgical repair [15,16].

The purpose of this retrospective analysis was to review all jersey finger type V injuries treated at an academic Level 1 trauma center over a period of 19 years to describe injury characteristics, primary radiological findings, treatment options and subsequent patient outcomes, as well as occurring complications. We aimed to evaluate potential analogies within the patient groups of type V injuries and reveal potential advantages of specific therapeutic approaches.

## 2. Materials and Methods

We ran a database query, retrieving all patients treated for a fracture of the distal phalanx between 2002 and 2020. All the patients consulted our academic Level 1 trauma center at least once. In total, 1580 patients with a fracture of the distal phalanx were treated in this period. All 1580 patients’ charts and their initial X-rays were reviewed for study inclusion by a single surgeon. A total of 44 patients with injuries matching the criteria for classification as jersey finger type Va and type Vb defined by Al-Quattan [6] were detected. Of these patients, 23 matched the classification criteria of jersey finger type Va and 21 those of jersey finger type Vb. All patients with a jersey finger type V were rechecked and confirmed by a second surgeon.

The primary examination of the patients was performed by the orthopedic trauma surgeon on duty; each primary X-ray was reviewed by an orthopedic trauma surgery-consultant additionally. The clinical follow-up was conducted in the outpatients’ clinic for hand injuries. The therapeutical approaches were determined by fellowship-trained hand surgeons, who also performed the surgical treatment if necessary. Indication for surgery included severe displacement between the avulsed, as well as fractured, fragments (>2 mm); evidence for retraction of the FDP tendon (which by definition is marked by a bony fragment in every jersey finger type V); the presence of rotational or angular malalignment; and DIP joint subluxation. The technique for surgery was designated by the treating hand surgeon.

All patients received a static finger splint for the injured finger for 4 weeks, regardless of treatment modality. Subsequently, the patients treated surgically received a DIP joint splint for another 2 weeks until removal of the K-wire and the Bunnell pull-out suture, if used. After removal of the hardware and splint, hand therapy was performed according to the flexor tendon rehabilitation protocol at our department. Periodic assessments of the ROM were conducted by using a goniometer. The median time of ambulatory care was 30 days (24–45 days).

All medical records of the patients, including the discharge summaries, as well as diagnostic, therapeutic and operative reports, together with radiographic, CT and MR images, were included in the evaluation of the patients for this study. All images were analyzed using an IMPAX workstation (Agfa, Ridgefield Park, NJ, USA). Due to calibration of magnification and distances of performed medical imaging, exact distance measurements were facilitated. All patients were analyzed in the corresponding jersey finger subtype group. Furthermore, we subdivided the patient cohort into two subgroups: (1) severe displaced (SD), matching at least one indication for surgery as mentioned above, and (2) minimally displaced (MD) injuries (Figure 2).

Statistical analysis was performed by using IBM SPSS Statistics Version 26, 64 bit. Normal distribution was assessed for every parameter by the analysis of histograms. Data of normally distributed parameters are presented as the mean and its standard deviation, and non-normally distributed data as the median and its interquartile range in round brackets. The functional outcome was transformed from a nominal scale into an ordinal scale for better comparison. The basis of the transformation was the range of motion. Patients with a loss of range of motion (ROM) of totally less than 10 degrees were classified as “excellent”, a loss of ROM of less than 30 degrees as “good” and a loss of ROM of less than 60 degrees as “fair”. A reduction in the ROM of more than 60 degrees was considered a poor outcome.

## 3. Results

Our retrospective data analysis revealed 44 patients who sustained a jersey finger type V injury. A total of 26 of these patients were male, 18 were female, and the mean age at the time of injury was 49 years (33–66 years of age).

In these 44 patients, direct blows were the most common mechanism of injury (23 cases), but also crush injuries of the distal phalanx (7 cases) and hyperextension traumas (6 cases) were reported. In four cases, the injury was related to a brawl and in two cases to an explosion. Two patients could not remember or specify the mechanism of injury. Overall, no compound avulsion fracture was observed.

In 23 cases, the injuries were associated with the left and in 21 cases with the right hand. Most frequently, the little finger was injured (16 cases), followed by the ring finger (13 cases). The middle and forefinger were injured in eight and seven cases, respectively.

The median time to follow-up was 30 days (24–45 days). Patients who underwent surgery were followed up for up to 6 months after surgical intervention.

### 3.1. Type Va

Of the 23 patients in the jersey finger type Va group, 7 patients showed a severely displaced injury (SD). However, only three of them eventually underwent surgery. The remaining four patients were lost to follow-up. One of them sickened before operative treatment. Another did not consent to surgery. Additionally, two patients were first diagnosed with an isolated transverse fracture of the distal phalanx and were treated with a splint. However, a bone-flake volar to the middle phalanx was subsequently noticed by the reviewing consultant orthopedic trauma surgeon, indicating a potential additional avulsion of the FDP tendon. Although the patients were immediately contacted and informed about the findings, they neither returned to our clinical department to complete diagnostics with further medical imaging nor attended any further appointment and were thus lost to follow-up.

The median primary bony displacement in patients in the SD subgroup was 0.9 mm (0.6–3.0 mm). Significant retraction (level of PIP joint) of the FDP tendon was observed in every patient. The functional outcome in cases who subsequently underwent surgery was poor in one patient and fair in another. One patient who presented at our department was in clinical care at another trauma center. He had already undergone surgery; however, he showed signs of pseudarthrosis when he presented at our clinic and had already received an appointment for revision surgery. He consulted our department only once; further patient history and functional outcome were not recorded. Surgery was performed using Bunnell pull-out sutures and K-wires in all patients (Figure 3).

In the remaining 16 patients in the MD subgroup, the median primary bony displacement in the other 16 patients with MD injuries was 0.85 mm (0.6–1.0 mm). No significant retraction of the FDP tendon was detected. No relevant secondary bony displacement was observed. Six patients were lost to follow-up, therefore no outcome was recorded. The reported functional outcome, corresponding to the ROM in the remaining 10 patients, was excellent in 6 patients and good in 4 patients (Table 1).

Overall, 3 of 20 patients showed signs of pseudarthrosis after completion of treatment, all of them in the transverse component of the avulsion fracture.

A total of 3 of 23 patients sustained this injury due to an enchondroma of the distal phalanx. One of the latter patients underwent surgery and achieved an excellent functional outcome after curettage and bone grafting. However, the fracture was only slightly displaced and did not match the indications for reconstruction stated by our study protocol; therefore, the surgical outcome was not taken into further consideration in this evaluation.

### 3.2. Type Vb

Of the 21 patients who sustained jersey finger type Vb injuries, 7 patients showed a severely displaced injury (SD). In the end, only three patients were treated surgically at our department. Among the other four, two refused surgery and were lost to follow-up, and one wished to be operated on in another clinic and was thereby lost to follow-up. The last one presented with a 3-week-old injury and rejected surgery. Furthermore, no patient with a jersey finger type Vb was missed in a primary examination.

The median primary bony displacement in patients with SD injuries was 5.8 mm (1.2–6.3 mm) and the median articular step-off 2.5 mm (0.9–3.9 mm). The functional outcome in cases who subsequently underwent surgery was poor in two and good in one patient. The patients with a poor outcome developed osteoarthritis grade 4 according to the Kellgren–Lawrence Score within 6 months, resulting in complete stiffness of the DIP joint. Their injuries were reconstructed with the help of a Bunnell pull-out suture and a K-wire in one patient and with three K-wires in the other. The patient receiving reconstruction with the Bunnell pull-out suture lost the ulnar part of his nail due to the refixation. The patient with a good outcome received reconstruction with a mini-cortical screw and a K-wire (Figure 4).

The median primary bony displacement in the 14 patients with MD injuries was 0.65 mm (0.5–1.025 mm). A median secondary bony displacement of 0.1 mm (0.1–1.3 mm) was observed. An articular step-off of 0.55 mm (0.4–0.8 mm) was observed, which extended about 0.1 mm (0.0–0.225 mm) during treatment. The reported functional outcome, corresponding to the ROM in these 14 patients, was excellent in 6 patients, good in 3 patients and fair in 1 patient. In four patients, no outcome was recorded due to loss to follow-up (Table 1).

Overall, 3 of 16 patients showed signs of pseudarthrosis after completion of treatment, all of them in the transverse component of the avulsion fracture, and 1 of 21 patients sustained this injury due to an enchondroma.

## 4. Discussion

Treatment of jersey finger type V is demanding, and the literature concerning therapy options and outcomes is scarce. Only a few case reports and case series can be found in the literature, reasoning out the integration of type V into the classification of jersey finger injuries and reporting on diagnostic approaches, repair techniques and subsequent outcomes.

In the case series of Al-Quattan [6], who was the first to describe jersey finger type V, all operated cases reached an almost unimpaired ROM after treatment without concomitant complications. The reconstruction of type Va was performed using only one 3–0 polypropylene pull-out suture; in cases of type Vb injuries, K-wires for DIP joint stabilization were added. Although we used the same technique for reconstruction in some patients, the achieved results were not as favorable. We also encountered one case with postoperative nail deformity, as this is a common complication when using pull-out sutures for reconstruction. The patient receiving reconstruction with three K-wires showed a worse radiological outcome, resulting in joint stiffness after a postoperative period of 4 months. Although this method of refixation showed sufficient results in the treatment of jersey finger type III injuries [17], it seems inappropriate for jersey finger type V injuries. Another fateful outcome was observed in the type Vb injury reconstructed with a Bunnell pull-out suture and one K-wire, resulting in severe osteoarthritis after 6 months. Refixations with these techniques seem to treat these severe injuries insufficiently.

Al-Quattan et al. [6] described their observed injuries as related to sports. However, they did not specify the mechanism of trauma, which resulted in the jersey finger type V.

Crowley et al. [9] reported on a jersey finger type Va injury in 2014 and simultaneously proposed an external fixator for treatment, as their patient also sustained an intraarticular fracture of the middle phalanx. The patient was injured by getting trapped in a motorized hedge trimmer, whereby he also sustained a ragged laceration at the level of the volar aspect of the DIP joint. They reconstructed the middle and distal phalanx by using seven K-wires for bone and joint reconstruction and a four-strand pull-out suture for tendon refixation. The active ROM in the DIP joint was 15 to 55 degrees after one year. They did not encounter any complications such as pin-site infections or nail growth aberrations.

Kang et al. [10] introduced a technique for the reconstruction of bony FDP tendon avulsions by using a miniplate in 2003. One of the five patients they reported on suffered from a jersey finger type II with a concurrent transverse fracture of the distal phalanx. However, this also matches the classification criteria of jersey finger type Va. They used a two-hole titanium miniplate, two 1.5 mm bicortical screws and two half-Kessler sutures for reconstruction. An early active motion protocol was started the day after surgery. Excellent results were achieved in all patients except one treated with this type of refixation. The patient with the jersey finger type Va injury only suffered from a 10-degree restriction in flexion and a loss of 3 kg grip strength compared to the uninjured site.

In 2018, Nho et al. [11] reported two type II jersey fingers. However, the published X-rays of their second case showed a concurrent minimally displaced transverse fracture of the distal phalanx as well. They performed reconstruction by using a single suture anchor. The authors reported a ROM of 0–10−70 after a postoperative period of three months. Although the possibility of refixating bony avulsions of the FDP tendon by using a single suture anchor and a specific suture technique was described in previous biomechanical studies, this technique was not tested and adapted for type V avulsions [18,19]. Furthermore, Nho et al. did not report the suture technique they used.

In 2011, Rizis and Mahoney [8] reported on one patient with a jersey finger type Vb injury, which they reconstructed by using a pull-out suture for fragment reduction and a K-wire placed in a longitudinal direction volar to the avulsed fragment for additional retention. No postoperative complications were observed, and an excellent ROM was achieved. However, limited information concerning treatment modalities and the functional outcome was provided in this case report.

Recently, Compton et al. challenged the need for acute repair of zone 1 FDP tendon injuries [16]. They found similar functional outcomes in patients undergoing FDP tendon repair compared to patients treated conservatively. However, in patients without FDP tendon repair, far fewer complications in terms of wound dehiscence, infections or secondary surgeries were observed. In our study, patients treated conservatively even had a more favorable outcome compared to those who underwent FDP tendon refixation. However, in our study, the therapy was determined by the severity of the injury; therefore, a direct comparison is not applicable. Furthermore, the study of Compton et al. did not involve bony FDP tendon avulsions.

In this present study, we could not confirm a functionally sufficient outcome in patients suffering from a jersey finger type V injury after reconstruction with established techniques using K-wires and pull-out sutures. Also, the reported complications of extracutaneous fixations should not be underestimated [12,20]. Therefore, new techniques like the miniplate refixation or the combination of screws, suture anchors and K-wires should be considered and evaluated for potential establishment in the surgical routine. Moreover, conservative treatment should be considered in minimally displaced avulsions without the presence of rotational or angular malalignment, as they showed good outcomes. Furthermore, we cannot confirm that the typical trauma mechanism of a jersey finger is also pathognomonic for type V jersey fingers. We rather observed direct blows, crush injuries and hyperextensions as predominant trauma mechanisms. Also, the distribution of injuries to each finger differs from the other types of jersey finger injuries, as they mainly affect the fourth finger, whereas we found rather an affection of the fifth finger in jersey finger type V [21,22]. However, within the two subgroups of jersey finger type V, radiological and clinical findings showed homogenic characteristics. Therefore, we consider these injury patterns worth being characterized on their own; however, their inclusion in the group of jersey finger injuries may be misleading.

Nevertheless, jersey finger type V is not as rare as usually presumed, as it accounts for 2.8% of the fractures of the distal phalanx in our collective (44 jersey fingers type V in 1580 distal phalanx fractures). Moreover, awareness of FDP tendon avulsions concomitant with transverse fractures of the distal phalanx should be increased to prevent misdiagnoses.

Limitations of this study are indicated by its retrospective character. Due to the involvement of various surgeons and their varying thoroughness in documentation, differences between the patient groups cannot be evaluated precisely. In the group of SD jersey finger V, not every patient underwent surgery; therefore, surgical outcomes could improve with a higher number of surgeries. Furthermore, a lot of patients were lost to follow-up. However, this is by far the largest number of conservatively as well as operatively treated patients reviewed in a study concerning this injury. Nonetheless, a low incidence of jersey finger type V leads to comparatively small study populations.

## 5. Conclusions

We were able to confirm homogeneity within the two subgroups of jersey finger type V. However, they differ greatly from jersey finger types I–IV concerning the trauma mechanism and affected finger; therefore, their inclusion in the group of jersey finger injuries should be reevaluated. Established surgical methods for refixation did not show sufficient outcomes, thus necessitating further investigation of innovative techniques. However, minimally displaced jersey finger type V injuries should be treated conservatively, as they achieve good functional outcomes.

## Figures and Tables

**Figure 1 jcm-13-06540-f001:**
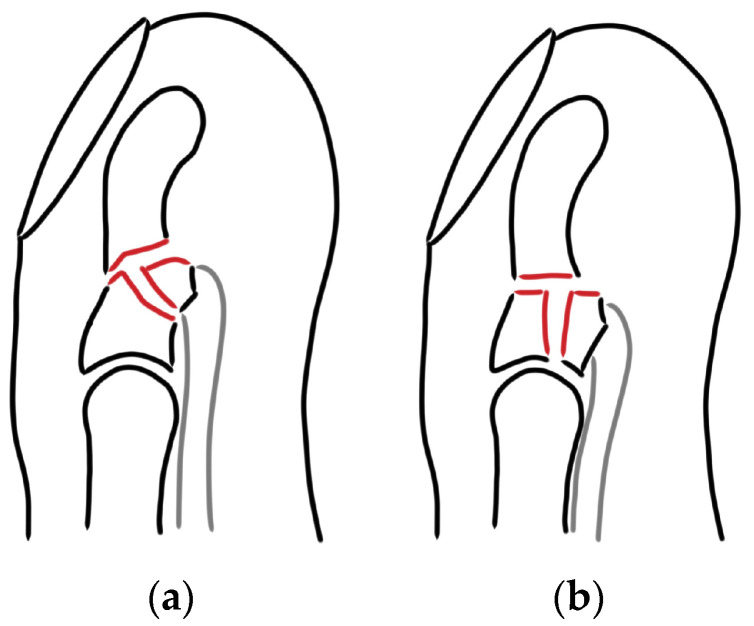
Schematic illustration of jersey finger type Va (**a**) and type Vb (**b**) injuries.

**Figure 2 jcm-13-06540-f002:**
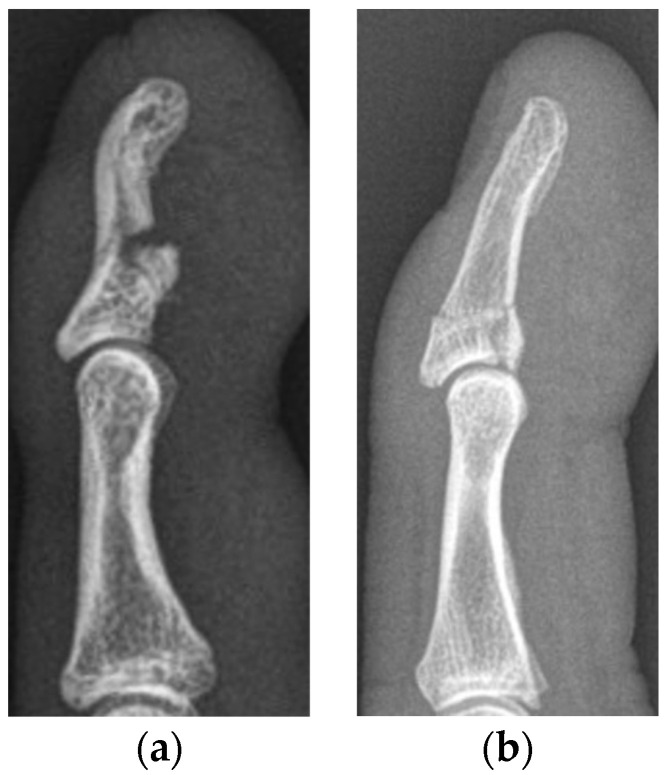
Lateral radiographs of a minimally displaced type Va (**a**) and type Vb (**b**) jersey finger, treated conservatively in our department.

**Figure 3 jcm-13-06540-f003:**
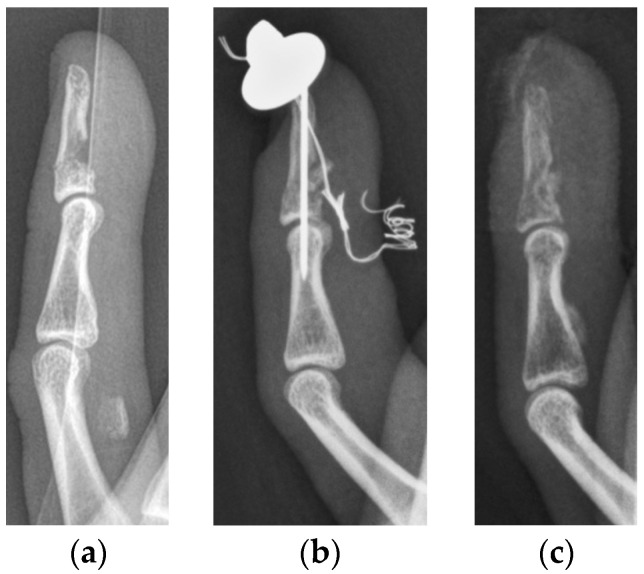
Lateral radiographs of a type Va jersey finger, with significant retraction of the FDP tendon together with a bony avulsion fragment (**a**). The same patient after surgical refixation using a K-wire and a Bunnell pull-out suture as proposed by Al-Quattan et al. [6] (**b**). Lateral radiographs after implant removal 6 weeks after surgery (**c**). The functional outcome of this patient was classified as fair.

**Figure 4 jcm-13-06540-f004:**
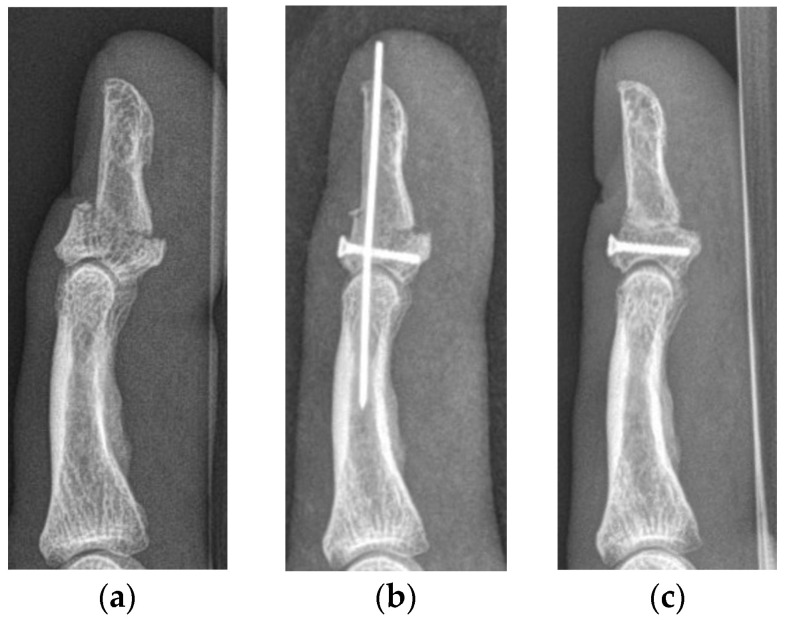
Lateral radiographs of a type Vb jersey finger, with severe displacement of the bony fragments and a resulting articular step-off (**a**). The same patient after surgical refixation using a mini-cortical screw and a K-wire (**b**). Lateral radiographs after K-wire removal 6 weeks after surgery (**c**). The functional outcome of this patient was classified as good.

**Table 1 jcm-13-06540-t001:** Functional outcome corresponding to the achieved ROM, differentiated between jersey finger type Va and Vb, as well as minimally displaced (MD) and severely displaced (SD) injuries. Patients with enchondroma included.

	Type Va-MD (*n* = 16)	Type Va-SD (*n* = 7)	Type Vb-MD (*n* = 14)	Type Vb-SD (*n* = 7)
Excellent	6	-	6	-
Good	4	-	3	1
Fair	-	1	1	-
Poor	-	1	-	2
Lost to follow-up	6	5	4	4

## Data Availability

The raw data supporting the conclusions of this article will be made available by the authors on request.

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
