# Peer review of "Characteristics and Therapy of Jersey Finger Type V Injuries at a Middle-European Level 1 Trauma Center—A Retrospective Data Analysis"

_jcm, 2024, doi:10.3390/jcm13216540_

Round 1

Reviewer 1 Report

Comments and Suggestions for Authors

This was an excellent article that I enjoyed reading and that I believe is worth publication. Congratulations for reporting characteristics and outcomes of this largest cohort of patients with type V Jersey fingers

1. Line 14 : « and » not « und »

2. Results: when reporting functional results of ROM, please include the median time to follow-up. Were these assessed at 30 days postop? 6 months? 1 year?

3. Please include in the limitations section that there were a lot of patients lost to follow-up

Author Response

Thank you very much for taking the time to review this manuscript. Please find the detailed responses below and the corresponding revisions/corrections highlighted in the re-submitted files.  

Comments 1. Line 14 : « and » not « und »

Response: Thank you for pointing this out. It has been corrected as quoted.

Comments 2. Results: when reporting functional results of ROM, please include the median time to follow-up. Were these assessed at 30 days postop? 6 months? 1 year?

Response: Thank you for the improvement, we clarified this by including the following sentence in the results section: p. 4, l. 133-134 “The median time to follow-up was 30 days (24 – 45 days). Patients who underwent surgery were followed up for up to 6 Months after surgical intervention.

Comments 3. Please include in the limitations section that there were a lot of patients lost to follow-up

Response: Thank you for pointing this out, we included this in the limitations section (p. 8, l. 301 “Furthermore, a lot of patients were lost to follow-up.”)

Reviewer 2 Report

Comments and Suggestions for Authors

Introduction: Enhance this section with recent information regarding this topic. How about the postoperative complications and outcome? How about the quality of life of these patients? 

Materials and Methods: Were the hand therapy protocols identical for all patients? Could they interfere with the general evolution of the affected finger?

Discussions: Sustain this sentence with more data: 'Nevertheless, jersey finger type V is not as rare as usually presumed, as it accounts 279 for 2,8 % of the fractures of the distal phalanx in our collective.'

References: Kindly add more data to support your ideas in this study (focus on articles published within the previous ten years timeframe) - Introduction and Discussion sections.

Author Response

Thank you very much for taking the time to review this manuscript. Please find the detailed responses below and the corresponding revisions/corrections highlighted in the re-submitted files.

Comments 1. Introduction: Enhance this section with recent information regarding this topic. How about the postoperative complications and outcome? How about the quality of life of these patients? 

Response: Thank you for your comment, accordingly, we included more recent information concerning postoperative complications and outcome in the introduction section (p. 2, l. 61-64 “Recent studies have reported high rates of complications in patients with FDP tendon repair in zone 1, in terms of osteomyelitis, wound dehiscence, chronic draining granuloma and the need for secondary surgeries, challenging the benefits of surgical repair (15,16).”)

Comments 2. Materials and Methods: Were the hand therapy protocols identical for all patients? Could they interfere with the general evolution of the affected finger?

Response: Thank you for pointing this out. Our patients received specialized therapy according to identical standardized hand therapy protocols. However, the adherence of patients may vary, this was not evaluated separately in this study. For better understanding we adapted the sentence (p. 3, l. 95-96 "After removal of hardware and splint, hand therapy was performed according to the flexor tendon rehabilitation protocol at our department.")

Comments 3. Discussions: Sustain this sentence with more data: 'Nevertheless, jersey finger type V is not as rare as usually presumed, as it accounts 279 for 2,8 % of the fractures of the distal phalanx in our collective.'

Response: Thank you for pointing out this vagueness, we included the fundamental data at the end of the sentence (p. 8, l. 294-295 “(44 jersey fingers type V in 1580 distal phalanx fractures)”)

Comments 4. References: Kindly add more data to support your ideas in this study (focus on articles published within the previous ten years timeframe) - Introduction and Discussion sections.

Response: Thank you for your comment, accordingly, we included more literature in the introduction section concerning postoperative complications and outcome (p. 2, l. 61-64 Recent studies have reported high rates of complications in patients with FDP tendon repair in zone 1, in terms of osteomyelitis, wound dehiscence, chronic draining granuloma and the need for secondary surgeries, challenging the benefits of surgical repair (15,16).) and discussed this in the discussion section (p. 7, l. 267-275 Recently, Compton et al. challenged the need for acute repair of zone 1 FDP tendon injuries (16). They found similar functional outcomes in patients undergoing FDP tendon repair compared to patients treated conservatively. However, in patients without FDP tendon repair far less complications in terms of wound dehiscence, infections or secondary surgeries were observed. In our study, patients treated conservatively even had a more favorable outcome compared to those who underwent FDP tendon refixation. However, in our study, the therapy was determined by the severity of the injury, therefore a direct comparison is not applicable. Furthermore, the study of Compton et al. did not involve bony FDP tendon avulsions.)